# Group 2 Innate Lymphoid Cells: Central Players in a Recurring Theme of Repair and Regeneration

**DOI:** 10.3390/ijms21041350

**Published:** 2020-02-17

**Authors:** Melina Messing, Sia Cecilia Jan-Abu, Kelly McNagny

**Affiliations:** 1Division of Experimental Medicine, Faculty of Medicine, University of British Columbia, The Biomedical Research Centre, 2222 Health Sciences Mall, Vancouver, BC V6T 2B9, Canada; mmessing@brc.ubc.ca; 2Department of Medical Genetics and School of Biomedical Engineering, Faculty of Applied Science and Faculty of Medicine, University of British Columbia, The Biomedical Research Centre, 2222 Health Sciences Mall, Vancouver, BC V6T 2B9, Canada; sia.jan-abu@ubc.ca

**Keywords:** Innate lymphoid cells, inflammation, repair, regeneration

## Abstract

Innate lymphoid cells (ILCs) are recently discovered innate counterparts to the well-established T helper cell subsets and are most abundant at barrier surfaces, where they participate in tissue homeostasis and inflammatory responses against invading pathogens. Group 2 innate lymphoid cells (ILC2s) share cytokine and transcription factor expression profiles with type-2 helper T cells and are primarily associated with immune responses against allergens and helminth infections. Emerging data, however, suggests that ILC2s are also key regulators in other inflammatory settings; both in a beneficial context, such as the establishment of neonatal immunity, tissue repair, and homeostasis, and in the context of pathological tissue damage and disease, such as fibrosis development. This review focuses on the interactions of ILC2s with stromal cells, eosinophils, macrophages, and T regulatory cells that are common to the different settings in which type-2 immunity has been explored. We further discuss how an understanding of these interactions can reveal new avenues of therapeutic tissue regeneration, where the role of ILC2s is yet to be fully established.

## 1. ILCs: A Brief Overview

In just over a decade, the discovery of innate lymphoid cells (ILCs) revealed a new branch of the innate immune system whose full functional relevance we are just beginning to appreciate. Due to their low frequency in normal tissues and the difficulty in distinguishing them due to their complex surface marker phenotype, ILCs remained undetected until recently but are now recognized as tissue-resident, cytokine-producing cells in a variety of organs. ILCs are divided into three main subsets (ILC1, ILC2, and ILC3) based on their surface markers, cytokine profiles, and functions (Table 1). Based on these features, ILC subsets closely resemble the T helper cell subsets (Th1, Th2, and Th17) found in the adaptive immune compartment, respectively, but ILCs lack surface expression of rearranged T cell receptors and antigen specificity [1].

Group 1 innate lymphoid cells (ILC1s) produce the typical type 1 cytokines interferon gamma (IFNγ) and tumor necrosis factor alpha (TNFα) and therefore participate in a type 1 immune response aimed at the clearance of intracellular bacterial and viral infections via macrophage recruitment and antigen specific Th1-mediated cell clearance [2]. ILC1s also respond to IL-12 and IL-18 signaling and rely on the t box transcription factor (Tbet/Tbx21) to generate a functional immune response [3]. Classic natural killer (NK) cells are grouped with ILC1s due to similarities in terms of cytokine production and transcription factor expression. However, functionally, NK cells are innate counter parts to CD8+ cytotoxic T cells due to the production of perforin and granzymes for targeted cell killing. NK cells are further distinguished by the expression of the transcription factor eomesodermin (EOMES) that is required for NK cell development.

ILC2s, which resemble Th2 cells, are primarily tissue resident and are prominent in lung, intestine, skin, and adipose tissue, where they participate in generating a type-2 immune response through the production of the cytokines IL-4, IL-5, IL-9, and IL-13 [4]. ILC2s rely on the transcription factors retinoic-acid-receptor-related orphan nuclear receptor alpha (RORα) and GATA binding protein 3 (GATA3) that are activated in response to IL-25, IL-33 and thymic stromal lymphopoietin (TSLP) signaling [5]. The downstream effects of ILC2 activation include B cell class switching, and subsequently IgG1 and IgE release, as well as the recruitment and degranulation of eosinophils, basophils, and mast cells. These are key features of an allergic response and lead to symptoms such as histamine release, mucus production, and muscle contraction. In the context of inflammation and infection, ILC2s may also produce amphiregulin (Areg), a member of the epidermal growth factor family of signaling factors, which facilitates the restoration of tissue integrity and homeostasis [6,7].

ILC3s, in contrast, are the innate counterpart to the Th17 immune cells whose responses are initiated by transforming growth factor beta (TGFβ), IL-1β, and IL-23 signaling and play important roles in the clearance of extracellular bacterial infections [8]. Additionally, ILC3s perform key functions in tissue homeostasis with prevalence in the gut mucosa. ILC3s can be identified based on the production of the cytokines IL-22 and IL-17 as well as the transcription factor retinoic-acid-receptor-related orphan nuclear receptor gamma t (RORγt) [9]. Like ILC2s, they express RORα, and recent data suggest that they also play a pathological role in the deposition of the fibrotic matrix in animal models for Crohn’s disease [10]. ILC3 subsets include those positive and negative for the natural cytotoxicity triggering receptor (NCR). NRC+ ILC3 only produce IL-22, while NCR- ILC3s produce both IL-22 and IL-17, as do lymphoid tissue-inducer (LTi) and this unusual ILC3 subset is particularly important for lymphoid structure development [11,12,13,14].

## 2. ILC2s: Key Interactions 

ILC2s are recognized as mediators of type-2 immune responses due to their release of type-2 cytokines and responsiveness to a number of damage-associated inflammatory signals through receptors for alarmins, prostaglandins, interferons, and cytokines [15,16,17]. Our understanding of ILC2s is becoming increasingly complex due to newly recognized subsets and subtle differences between ILC2s isolated from distinct tissue types and at different stages of inflammatory responses. However, there are several central and recurring tenets of ILC2 activation, interactions, and regulation across different developmental time points, tissues, and inflammatory conditions. These include ILC2-stroma crosstalk through the stromal production of IL-33, ILC2-mediated eosinophil activation via their production of IL-5 and IL-13, the ILC2-driven maintenance of M2 macrophages via IL-4 and IL-13, and the repression of ILC2s and type-2 immunity by IL-10 and T regulatory cells. Each of these aspects is introduced in the following and illustrated in Figure 1.

### 2.1. Stroma and IL-33 

Classically, stroma is considered to be the structural support of tissues and organs as well as a source of trophic factors for tissue homeostasis and repair [18]. Stromal cells are highly active in communication with other local cells and release signaling molecules at steady state (in response to metabolic or mechanical changes) as well as under an inflammatory condition and in response to tissue damage. IL-33 is stored in its active form in stromal cells (e.g., lung parenchymal cells) and is primarily released as an alarmin from damaged or dying cells [19]. The IL-33 receptor IL1RL1 (T1/ST2) is found on several immune cells, including ILC2s. As a member of the IL-1 family, the binding of IL-33 to ST2 and the co-receptor IL-1RAcP leads to Nuclear factor kappa-light chain-enhancer of activated B cells (Nf-κB) activation and GATA3 phosphorylation [20]. GATA3 interacts with the regulatory regions of type-2 cytokine genes, including IL-4, IL-5, and IL-13 to initiate their transcription. Therefore, IL-33 is a potent activator of ILC2s and a critical stepping-stone to a functional type-2 immune response [21]. The stroma and ILC2s also interact bi-directionally since ILC2s have been shown to directly modulate stromal cells through cytokines and growth factors. ILC2-derived IL-13 can induce goblet and tuft cell differentiation in the intestine while ILC2-derived Areg enables epithelial repair both in the intestine and the lungs following damage and inflammation [6,7,22,23]. 

### 2.2. Eosinophils

The alarmin-mediated ILC2 proliferation and expression of type-2 cytokines is essential for the recruitment and maintenance of eosinophils that are widely recognized as important inflammatory cells in allergic responses including asthma for which eosinophilia is one of the hallmarks [24]. ILC2-derived IL-5 is critical for eosinophil survival and function. It signals through the IL-5 receptor on eosinophils, which activates both Janus kinase (JAK)/Signal transducers and activator of transcription (STAT) as well as Ras/mitogen activated protein kinase (MAPK) signaling pathways. In addition to IL-5, eosinophil recruitment and subsequent eosinophilia rely on ILC2 derived IL-13 and stroma derived eotaxin—an essential eosinophil chemokine [25].

Eosinophil specific compounds, such as a major basic protein (MBP) or eosinophil peroxidase (EPO) are released following degranulation in response to an inflammatory environment and are critical for the clearance of parasitic infections [26]. Additionally, eosinophils are an important source of type-2 cytokines, such as IL-4 and IL-13, and thereby reinforce a type-2 immune response. At steady state, eosinophils have been implicated in the regulation of metabolic cycling as well as B cell survival [27,28].

### 2.3. M2 Macrophages

ILC2- and eosinophil-derived cytokines further determine the inflammatory phenotype of macrophages, which are likewise recruited in response to inflammatory stimuli. Unlike the IFNγ- and TNFα-dominated environment of a Th1 response that leads to classically activated (M1) pro-inflammatory macrophages, eosinophil- and ILC2-elaborated IL-4 and IL-13 favor transition to an M2 phenotype that is considered anti-inflammatory [29,30]. IL-4 and IL-13 share the IL-4Rα subunit, but IL-4 requires dimerization with the common gamma chain receptor subunit while IL-13 relies on the dimerization of IL-4Rα with IL-13R1a. Both cytokines activate JAK/STAT6 signaling and the downstream production of M2 specific proteins, such as arginase-1 and the mannose receptor, CD206. M2 macrophages have a reparative and resolving phenotype resulting in the production of growth factors such as vascular endothelial growth factor (VEGF) and TGFβ but also anti-inflammatory cytokines such as IL-10. This state of macrophage activation is necessary for repair following tissue damage and, in many cases, is also necessary for normal tissue function for which a pro-inflammatory environment would be detrimental [31]. However, prolonged M2 survival and activation can lead to excess TGFβ accumulation, which in turn can lead to repetitive fibroblast activation, pathological collagen, and other extracellular matrix deposition.

### 2.4. T Regulatory Cells

T regulatory cells (Tregs) are one subtype of CD4+ T helper cells and play essential roles in the control and resolution of inflammation. In addition to CD4 expression, Tregs can be identified by the expression of the transcription factor Forkhead box P3 (FOXP3), as well as the beta chain of the IL-2 receptor (CD25). Treg subsets include thymus-derived natural Tregs (nTregs) or induced Tregs (iTregs), in which case they arise in the periphery in response to IL-2 and TGFβ signaling [32,33]. In Tregs, IL-2 signals through the IL-2 receptor and activates the JAK/STAT pathway while TGFβ binds to TGFβRII, which activates TGFβRI and enables the translocation of Smad proteins to the nucleus. These signaling events result in the elevated production of FOXP3 and subsequently TGFβ and IL-10. IL-10, in turn, is a potent anti-inflammatory cytokine that exerts its function on many different cell types, including macrophages as well as ILC2s, which both express the IL-10 receptor [34]. In addition to dampening ILC2 proliferation and cytokine production via IL-10, Tregs also inhibit ILC2s through direct inducible T-cell co-stimulator- (ICOS) mediated cell–cell contact. This is likely complemented by their competitive sequestration of IL-2 and IL-33 since Tregs express the receptors for both of these cytokines [35,36,37]. Stroma-derived IL-33 can therefore also directly act on Tregs and has been shown to promote both Treg accumulation and differentiation [38].

## 3. ILC2s and Type-2 Immunity: Same Players, Different Settings 

The bulk of our understanding of ILC2 characteristics and functions is based on evidence from mouse studies due to: (1) the ready availability of solid tissues for analyses, (2) the abundance of models of inflammatory disease, and (3) the easy access to mice with genetic lesions in ILC function [15]. These models have revealed ILC2 involvement in many different settings, both during homeostasis and in inflammatory conditions. In the following, we explore the common thread across these different settings with a focus on the key interactions referenced above.

### 3.1. ILC2s in Metabolic Homeostasis 

Metabolic processes, such as glucose metabolism, rely on a close interplay with immune cells that maintain insulin sensitivity and allow for optimal nutrient processing at metabolic sites, such as white adipose tissue. ILC2s reside in adipose tissue and their proliferation/activation is triggered by the continuous release of IL-33 at steady state. Adipose tissue resident multipotent stromal cells (MSCs) are one identified source of IL-33 and further promote ILC2 proliferation through direct cell–cell contact [39]. ILC2s, in turn, sustain eosinophils via IL-5 secretion and indirectly support eosinophil recruitment and maintenance through IL-4 and IL-13, which promote the release of eotaxin from MSCs [40]. Obesity disrupts ILC2 responsiveness both in mice and humans [41]. Under lean conditions, eosinophils, together with ILC2s, maintain macrophages in an M2 state through the release of IL-4 and IL-13. In the absence of eosinophils, mice are more prone to obesity and insulin resistance while eosinophilia enhances insulin sensitivity [31,40]. The state of macrophage activation is a critical indicator of the metabolic health of adipose tissue and therefore the close interplay between ILC2s and eosinophils with the stroma to promote M2s is essential [42]. Mice fed with a high fat diet show a disruption of the type-2 environment, increased numbers of pro-inflammatory macrophages and cytokines, weight gain, and insulin resistance. This can be attributed to a reduction of Treg-derived IL-10 in adipose tissue that, in lean mice, controls adipose tissue inflammation. Adipose tissue Tregs express ST2 and treatment of obese mice with IL-33 reverses tissue inflammation and insulin resistance (Figure 2) [37,43].

Beyond adipose tissue, ILC2s also sustain the cycling of eosinophils in the periphery through the constitutive release of low levels of IL-5, which changes in response to nutrient intake [27]. The nutrient sensitivity of ILC2s is enabled through the expression of the vasoactive intestinal peptide (VIP) receptor VIPR2/VPAC2 that makes ILC2s responsive to this neuropeptide and thereby positions ILC2s as central players of metabolism. Additionally, while ILC3s are significantly reduced in the intestine of vitamin-A-deficient mice or mice treated with a retinoic acid inhibitor, IL-13 producing ILC2s expand under these conditions [44]. This change in tissue resident ILC frequencies in response to malnutrition increases both the susceptibility of mice to bacterial infections and the protection against helminth infection due to the elevated levels of IL-13. This increase in IL-13 is enabled by the metabolic adaptation of ILC2s to utilization of fatty acid oxidation (FAO) in a nutrient deprived environment, and the blockage of FAO diminishes IL-13 production [44,45]. Malnutrition does not result in the elevation of other ILC2-derived cytokines, likely due to the essential nature of IL-13 for barrier protection (Figure 2).

### 3.2. ILC2s in Neonatal Immunity

In the neonate, the immune system is skewed towards a type-2 environment, likely to minimize type-1 induced tissue damage that could result in premature labor and to allow for the establishment of tolerance [46]. In mice, ILC2s seed the lung approximately 10 days after birth, leading to a peak frequency three times higher than their numbers in adult lungs [47]. This response appears to be linked to a tightly orchestrated release of IL-33 derived from epithelial cells just prior to ILC2 recruitment. This IL-33 release is attributed to the stress of first-time exposure to inhaled air. Interestingly, it occurs independently of parasitic or allergic triggers, and alarmin levels remain well below those usually seen in an adult inflammatory response. Neonatal ILC2s not only proliferate but also produce large amounts of IL-5 and IL-13 in response to IL-33. A single exposure to an allergic trigger results in a robust type-2 response that greatly exceeds ILC2 responsiveness in the adult [47,48]. Even without such a trigger, ILC2s induce the recruitment of eosinophils to the neonatal lung through the production of IL-5, which peaks at postnatal day 11. At this point, eosinophils exceed the numbers of most other innate immune cells, including macrophages, neutrophils, and dendritic cells. In the RORα staggered mice (RORα^sg/sg^), which lack ILC2s, eosinophil recruitment to the lung is greatly reduced [47,49]. While eosinophil-mediated tissue remodeling is typically associated with disease, eosinophils may contribute to airway remodeling during development to allow for lung maturation [50]. 

The increase in ILC2 numbers in normal mice further coincides with macrophage expansion in the lung. ILC2-derived IL-13 maintains the M2 state of macrophages, which contributes important anti-inflammatory properties to protect the lung from detrimental infections [51]. Therefore, low-level IL-33 release, together with ILC2 activation, has a unique and essential role in initiating and maintaining a type-2 biased environment during early life. Exposure to allergens during this early window may shape responses to future allergic exposures experienced later in life. To control an exacerbated type-2 response and allow for the establishment of a balanced immune environment, the development of the lung microbiota and the subsequent establishment of a T regulatory cell subset are critical. In the neonate, Helios+ Tregs are present but cannot suppress a type-2 immune response [52]. Only with the development of the lung microbiota is a Helios- Treg population established, which greatly reduces allergen susceptibility of the developing lung and is therefore key in shaping the immune environment that then persists in the adult. Thus, already in early life, ILC2s can be placed at the center of the stroma—type-2 immune cell interactions that are necessary for healthy development but also determine vulnerability of infants to allergic disease (Figure 3).

### 3.3. ILC2s in Allergic Lung Inflammation

Inflammation is the natural defense against a host of harmful agents, including invading foreign pathogens, allergens, and toxins or, alternatively, those which arise through internally triggered signals in autoimmunity and cancer. As discussed above, a type-2 immune environment may occur at steady state to allow for normal tissue development or function. However, in the adult lung, ILC2s are recognized as part of a type-2 immune response against large parasites and allergens (Figure 3). The central role of ILC2 interactions with the stroma, eosinophils, macrophages, and T cells is well defined in the context of allergic disease, which is recognized as a modern, global health concern with only limited options for treatment, especially in severe cases of allergic asthma [53]. Peripheral blood of chronic asthmatic patients shows elevated levels of IL-5- and IL-13- producing ILC2s, which is paralleled in patient airways by increases in IL-33 and eosinophilia [54,55,56].

Mouse models of allergic disease that involve the administration of protease allergens, such as papain or house dust mite (HDM), confirm the critical role of ILC2s for the establishment of the type-2 inflammatory response. Mice deficient in ILC2s show greatly attenuated inflammation following intranasal allergen treatment. Intriguingly, ILC2s appear to be dispensable if the allergen is administered systemically or in the case of triggers that do not elicit a type-2 immune response [57,58]. Furthermore, this response occurs upstream of T and B cell, which require ILC2 cytokine production to elicit an adaptive type-2 immune response [59,60]. As in the neonatal lung, IL-33 and other alarmins trigger ILC2-derived IL-5 and IL-13 production, which are critical for eosinophil recruitment. However, in the absence of eosinophils, ILC2 proliferation is impaired, suggesting that ILC2s also require eosinophils for their function [61]. In summary, ILC2s play a critical role in priming and perpetuating a Th2 response to local insults but are dispensable and can be circumvented in instances where antigens are delivered systemically (Figure 1 and Figure 3) [58].

Beyond immediate inflammation, chronic asthma occurs due to a feedback circuit established by ILC2s that respond to epithelial IL-33 and produce IL-13 that feeds back to the epithelium and enhances IL-33 production as well as IL-33 receptor expression, resulting in persistent type-2 cytokine production [54]. Beyond this feedback loop, ILC2-derived IL-13 also directly destabilizes lung epithelial barrier integrity, leading to compromised lung structure and function [62,63]. Additionally, ILC2s gain a memory phenotype in response to allergic triggers, which enables a more potent inflammatory response in the case of subsequent exposure to allergens or alarmins [64]. As in other tissues, ILC2-derived cytokines maintain macrophages in an M2 state during lung inflammation and also release arginase-1 that acts on M2s, which in turn contribute additional cytokines, induce collagen deposition, and prevent type-1 inflammation [65,66].

The resolution of a type-2 allergic immune response is mediated by iTregs, which are critical to the prevention of long-term pathological consequences, including the development of chronic allergic asthma [33,34,35]. Resolving factors such as the lipid mediator maresin-1 trigger iTreg accumulation, which then suppress IL-5 and IL-13 production by ILC2s via IL-10 and TGFβ and promote the production of Areg [67]. Moreover, direct cell-cell contact through the interaction of ICOS with its ligand ICOSL is required for effective suppression of ILC2 mediated inflammation. iTreg maintenance is dependent on IL-2 availability and a reduction in iTregs, or the disruption of ICOS signaling is associated with more severe and persistent lung inflammation [34,68].

### 3.4. ILC2s in tissue fibrosis

Tissue stability, flexibility, and integrity depend on stromal cells (including fibroblasts and other structural cells) and the complex network of proteoglycans and matrix proteins that comprise the extracellular matrix [69]. The stroma provides both structural support as well as environmental cues that are important for tissue homeostasis and immune cell modulation during tissue damage. Extracellular matrix fragments can be sensed directly by immune cells and promote or dampen inflammation, while fibroblasts replace or modify this matrix in response to injury [70,71]. Inflammatory cells, including neutrophils and macrophages that are recruited to the site of injury, drive this process. In the case of an acute insult, inflammation is essential to promote tissue resident progenitor cell proliferation, migration, and repair, which quickly subside upon restoration of tissue integrity and function. If inflammation persists, however, as is seen in response to repetitive injury, chronic infection, or in autoimmune or auto-inflammatory disease, excess matrix is repetitively produced. This leads to pathological tissue fibrosis and scarring, instead of healthy and functional tissue production [69]. While TGFβ has been established as an essential mediator of fibrosis, IL-13 is also recognized to play a role both in a TGFβ-dependent manner, as shown in the context of colitis but also through an independent mechanism, as demonstrated in liver fibrosis [72,73,74]. Due to their direct interaction with stromal cells (e.g., sensing IL-33 in adipose tissue) and their production of IL-13 both at steady state and during inflammation, ILC2s are, potentially, key mediators of fibrosis development and are implicated in several fibrotic conditions [75].

#### 3.4.1. Lung Fibrosis

Idiopathic pulmonary fibrosis (IPF) is the most common type of lung fibrosis, which arises in the lung interstitium and affects tissue function. There are currently no available treatment options for this disease that leads to significant impacts on quality of life, as breathing is progressively restricted, and often results in lethality [76].

IL-33 and IL-25 levels are elevated in bronchoalveolar lavage fluid (BALF) samples of IPF patients, which is mirrored by an increase in ILC2s in both BALF and lung tissue [77,78]. The IL-33/ST2 axis in the context of ILC2s is further implicated in several studies that explore a bleomycin-induced mouse model of IPF [78,79,80]. ST2-deficient mice show decreased lung inflammation and attenuated fibrosis in response to bleomycin treatment both in the case of systemic ST2 gene inactivation as well as more selective ST2 gene inactivation in hematopoietic cells. In both cases, ILC2 expansion after damage is attenuated, while the adoptive transfer of ILC2s leads to an exacerbated phenotype. This exacerbated phenotype can be replicated via the administration of recombinant IL-33 and prevented via neutralizing anti-IL-33 antibodies [79,81]. In bleomycin-treated lungs, full-length IL-33 (flIL-33) is released from both alveolar macrophages and stromal cells and is processed to a middle-length form (mlIL-33) due to elevated levels of neutrophil proteases. mlIL-33 triggers IL-13 production by acting on both ILC2s and macrophages. This elevated IL-13, in turn, triggers macrophage polarization towards an M2 phenotype as well as IL-13 and TGFβ production (Figure 3). Fibroblasts in the lung subsequently respond to IL-13 and TGFβ by producing excess collagen that, over time, replaces the high functioning elastic lung tissue with an inelastic fibrotic matrix [81]. 

In contrast to ST2 deletion, mice lacking ICOS are ***more*** susceptible to bleomycin-induced injury and exhibit increased vascular leakage and mortality but fail to show increased fibrosis development [82]. ICOS+ ILC2s are known to expand in response to bleomycin damage and ICOS-deficient mice lack this expansion as well as the production of IL-5, but not IL-13. IL-5 administration in trans protects against bleomycin-induced mortality in ICOS-deficient mice and it is intriguing that IPF patients exhibit a reduced frequency of ICOS+ ILC2s [82]. These observations raise the possibility that ILC2 subsets differentially contribute to disease progression: ST2+ ILC2s may contribute to fibrosis development, while ICOS+ ILC2s may protect against lung damage. Alternatively, they may reflect temporal differences in the roles played by ILC2s at different stages of the development of fibrosis. Bleomycin-induced lung fibrosis is well known to pass through three distinct phases: 1) hematopoietic driven acute inflammation, 2) a slightly delayed acute vascular leak mediated by the endothelium, and finally 3) a fibrotic phase mediated by stromal-dependent fibroblast proliferation and maturation [83]. Thus, the ability of IL-5 to dampen mortality mediated by ICOS deficiency could reflect an upstream hematopoietic effect prior to the subsequent phase of vascular leak. The protective nature of ICOS+ IL-5+ ILC2s matches the finding that in the bleomycin mouse model, eosinophils are elevated but the absence of eosinophils does not impact fibrosis development [84], however, increased eosinophil numbers are also linked to increased disease severity in IPF patients [85]. 

Interestingly, collagen deposition in bleomycin-damaged mice increases following treatment with an anti–IL-2 antibody and mouse recombinant IL-2 protein complex (IL-2C) that has been shown to expand Tregs [86]. Additionally, IL-2C treatment favors a type-2 immune environment, possibly due to bleomycin-induced phenotypic Treg changes and the suppression of type-1 cytokines that counteract an exacerbated type-2 immune response [87]. 

#### 3.4.2. Liver Fibrosis

The liver is one of the few tissues in the body that has the remarkable ability to fully regenerate following injury and even following partial organ removal. Nevertheless, infections or chronic damage can lead to the development of liver fibrosis and cirrhosis—an end stage form of liver fibrosis, leading to liver failure and necessitating a liver transplantation [88].

Several stromal cell types, including epithelial cells and hepatic stellate cells, are known sources of IL-33 in the liver and, moreover, hepatic stellate cells are also the major source of excess collagen following their transition into myofibroblasts under inflammatory conditions [89]. IL-33 expression is increased in human fibrotic livers as well as in independent mouse models of liver fibrosis that are based on the administration of the toxins thioacetamide (TAA) or carbon tetrachloride (CCL_4_) [90,91]. Mice deficient in IL-33 as well as IL-13 and IL-4Ra show significantly reduced levels of fibrosis, likely indicating the involvement of IL-13 signaling downstream of IL-33 activation [74]. ILC2s are known to expand in response to IL-33 in liver fibrosis and are a major source of IL-13, which causes hepatic stellate cell activation independently of adaptive immunity (fibrosis is not attenuated in Rag deficient mice) [91]. The complete interaction of ILC2s with eosinophils, M2s and Tregs in the context of liver fibrosis are not fully understood but since ILC2s are implicated as central players in hepatic fibrosis it is likely that eosinophils as well as M2 macrophages participate as sources of IL-13 and perhaps TGFβ. Since Tregs commonly reduce inflammation, Treg responses may also be perturbed and this may be an additional mechanism behind the development of hepatic fibrosis. These aspects call for a more detailed evaluation of ILC2s and type-2 immune responses in the context of liver disease.

### 3.5. New Avenues: ILC2s in Tissue Regeneration

The recurring theme of ILC2 interactions in tissue inflammation and repair in distinct settings may provide novel insights into characteristics and functions in tissues and processes that are yet to be explored. As discussed above, the role of ILC2s has been explored in neonatal development, tissue homeostasis, inflammation/repair and fibrosis and, most extensively, in the immune response during infection and inflammation in barrier organs. Based on the precedents set by these studies, one could anticipate additional roles for ILC2s in the repair, regeneration, and homeostasis of non-barrier organs. A tissue that is very well known for its regenerative abilities is the skeletal muscle, which in healthy individuals fully restores muscle structure and function in response to acute, sterile damage that is often experienced in response to exercise. This efficient and complete regeneration is attributed to a well-orchestrated immune response and interplay of immune and muscle resident cells (Figure 4) [92,93].

Similar to wound healing, muscle regeneration moves through different stages of immune activation that, immediately after damage, is dominated by a type-1 pro-inflammatory cytokine response. Over time, this transitions to a type-2 environment that is thought to enhance repair. In this temporal sequence, pro-inflammatory cytokines, primarily IFNγ and TNFα, are released by neutrophils and macrophages. This enables satellite cell (muscle stem cell) activation, migration, and proliferation, which generate the pool of muscle progenitors (MPs) that eventually form new muscle fibers. In this early phase, macrophages clear debris and necrotic cells through phagocytosis [94,95,96]. However, the persistence of pro-inflammatory cytokines prevents muscle cell differentiation and therefore a change in immune environment is required for the recruited macrophages to transition towards a pro-repair phenotype [97]. A number of different events have been explored as the possible triggers for this phenotype switch, including intrinsic metabolic changes as a result of phagocytosis, differential signaling by lipid mediators due to lipid mediator class switching, as well as a change in cytokine environment [96,98,99]. Eosinophils are known to enter the muscle following damage and are a major source of IL-4 and IL-13 at the time of macrophage phenotypic transition [100]. These cytokines signal through the IL-4R expressed on a muscle resident population of mesechymal progenitors, which have been dubbed fibro/adipo progenitors (FAPs). FAPs support the repair of the basement membrane during normal muscle regeneration. Under chronic inflammatory conditions, FAPs differentiate into adipocytes and fibroblast and mediate fibrosis development [97,101]. In normal muscle regeneration, IL-4/IL-13 signaling prevents the differentiation of FAPs into adipocytes and enhances their release of trophic factors to support muscle fiber development [100]. Since eosinophil-derived cytokines are known to trigger and maintain an M2 phenotype in other settings [31], it is likely that eosinophils are recruited to the muscle at this critical time during regeneration to either initiate or support macrophage phenotypic switching in addition to their established interactions with FAPs. However, the signaling events that lead to eosinophil recruitment are currently not established. In contrast, it is known that, similar to stromal cells in adipose tissue, FAP-like cells release IL-33 at steady state (triggered by muscle movement), which is essential to the maintenance of muscle resident Tregs and the enhancement of Treg accumulation after damage [102]. Tregs are an important source of IL-10 as well as Areg during muscle regeneration, and it has been shown that the release of this IL-10 is impaired upon Treg depletion [103]. Macrophage phenotype switching requires IL-10 and therefore the loss of IL-10 prolongs inflammation and impairs muscle regeneration [104,105]. Collectively, these suggest that the key components of a type-2 immune response, including stroma-derived IL-33, eosinophil recruitment and IL-4/IL-13 signaling, and M2 and Treg accumulation, are present in damaged muscle tissue. In aggregate, these reveal a strong parallel to type-2 interactions observed in other settings. Therefore, it is likely that ILC2s are resident in skeletal muscle, activated by increased levels of IL-33 as a result of FAP proliferation, which in turn results in IL-5-dependent eosinophil recruitment and IL-4/IL-13-dependent macrophage phenotype switching. This type-2 response is likely resolved by IL-33 dependent Tregs that inhibit ILC2 activation via IL-10 signaling and possibly other mechanisms, including direct cell–cell contact or competition for essential cytokines. Accordingly, it is tempting to speculate that ILC2s may further contribute directly to muscle repair through the release of Areg since ILC2s contribute in this manner to the restoration of tissue homeostasis in other settings [6].

Chronic muscle inflammation is associated with muscular dystrophies and leads to fibrosis at the expense of muscle regeneration. In this setting, the evaluation of ILC2 function may be particularly important in revealing the significance of innate cell-driven type-2 immunity in disease development and resistance [106]. Intriguingly, eosinophils have been implicated in having a deleterious role in a mouse model of Duchenne Muscular Dystrophy (DMD) where chronic inflammation triggers eosinophil degranulation and the release of major basic protein (MBP) that contributes to chronic muscle damage [107]. In contrast, as demonstrated recently, neither eosinophil absence nor overexpression in dystrophic mice is associated with a difference in pathology [108]. On the other hand, the expansion of Tregs via IL-2C treatment significantly decreases muscle fiber damage in dystrophic mice due to elevated IL-10 production that limits type-1 inflammation [105,109]. The balance between pro-inflammatory and anti-inflammatory macrophages is particularly interesting since muscle fiber damage can be largely attributed to a type-1 immune environment but subsequent fibrosis development is driven by M2 macrophage-derived excess TGFβ [110]. However, as discussed above, both types of macrophages serve critical functions for muscle regeneration and therefore the full ablation of one type or one cytokine may not support long-term recovery of damaged or dystrophic muscle [97,111]. Therefore, the subtle manipulation of innate responses and macrophage phenotypes rather than the ablation of inflammatory cell types may be a more suitable treatment approach. Such an ILC2- and innate-response-focused approach could prove pivotal in extending the health and life span of individuals with these syndromes.

## 4. Concluding Remarks

ILC2s are central players of type-2 immunity, found across different tissues and in the context of different settings, including neonatal immunity, adult homeostasis, inflammation, and fibrosis. Key ILC2 interactions with stromal cells, eosinophils, macrophages, and T cells have immerged as the common thread that enables a comparison of ILC2 function between these different time points, tissues, and conditions. Understanding these fundamental interactions and the underlying cause of their repetitive nature despite variable circumstances has the potential to explain the contrasting contributions of type-2 immunity to healthy metabolic processes or protection against parasites compared to tissue destruction through chronic inflammation and fibrosis. Finally, as exemplified with the example of tissue regeneration, this common pattern can guide the way to previously unexplored territory, where ILC2s may play an essential role—beneficial or detrimental—and may not only add to our understanding of immunological mechanisms but also point to new therapeutic targets.

## Figures and Tables

**Figure 1 ijms-21-01350-f001:**
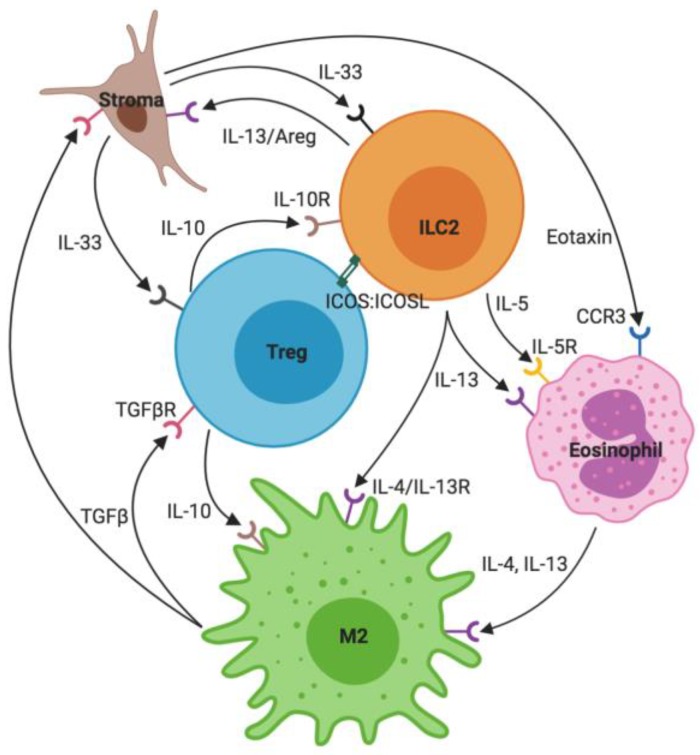
Key Group 2 innate lymphoid cells (ILC2) interactions. ILC2s are activated by the IL-33 released from stromal cells and produce mainly IL-5 and IL-13. These cytokines maintain eosinophils, which, together with ILC2s, in turn maintain an M2 macrophage phenotype. M2 macrophages release transforming growth factor beta (TGFβ), which both acts on stromal cells as well as Tregs. Tregs control a type-2 environment through IL-10 and inducible T-cell co-stimulator (ICOS) signaling.

**Figure 2 ijms-21-01350-f002:**
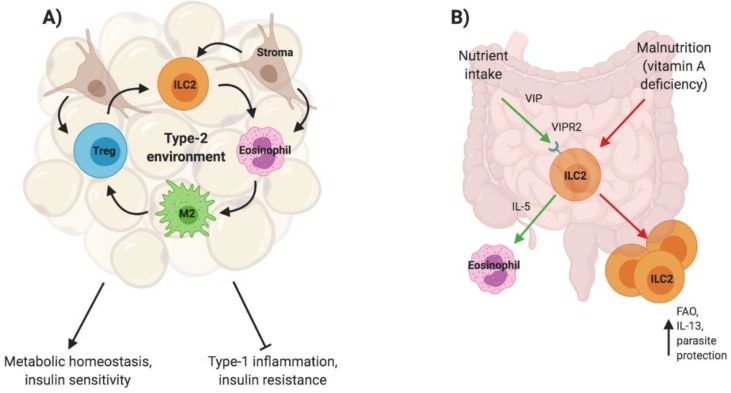
ILC2s in metabolic homeostasis. ILC2s maintain a type-2 environment in adipose tissue that is critical for a healthy metabolism (**A**). ILC2s are also found in the intestine and respond to nutrient intake to sustain eosinophil cycling. The adaptation to malnutrition includes the expansion of the ILC2 population and increased IL-13 production due to fatty acid oxidation (FAO) (**B**).

**Figure 3 ijms-21-01350-f003:**
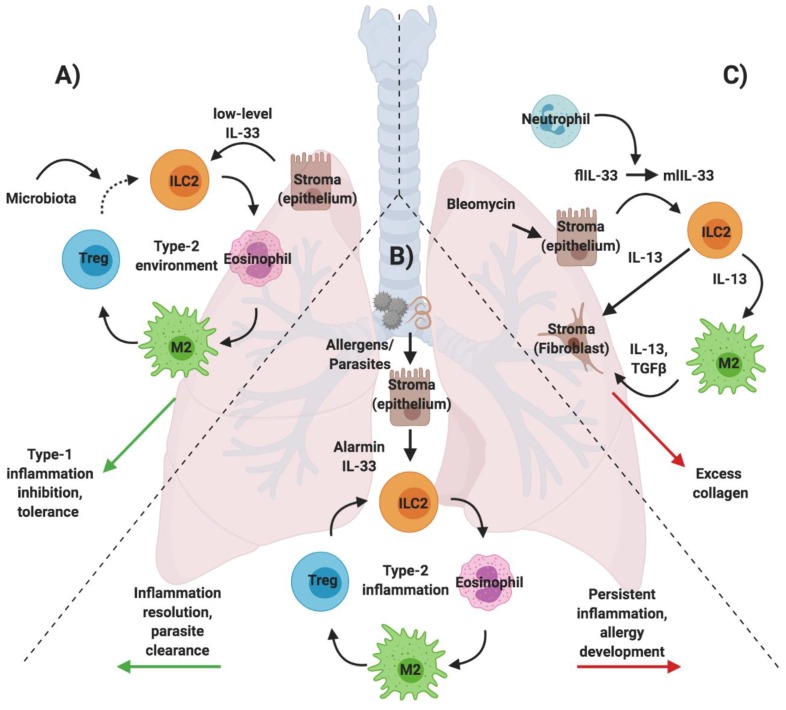
Key ILC2 interactions in lung development, inflammation, and fibrosis. During early development, low-level IL-33 released from epithelial cells triggers a type-2 environment. Lung microbiota colonization at later stages enables the Treg-mediated restraint of type-2 immune interactions (**A**). A type-2 immune response can be induced by allergens and parasites, which is either resolved quickly or may develop into allergic disease (**B**). Lung fibrosis is triggered by IL-33-associated ILC2 proliferation and the secretion of IL-13 that maintains an M2 macrophage phenotype, which, together with ILC2s, acts on fibroblast to produce excess collagen (**C**).

**Figure 4 ijms-21-01350-f004:**
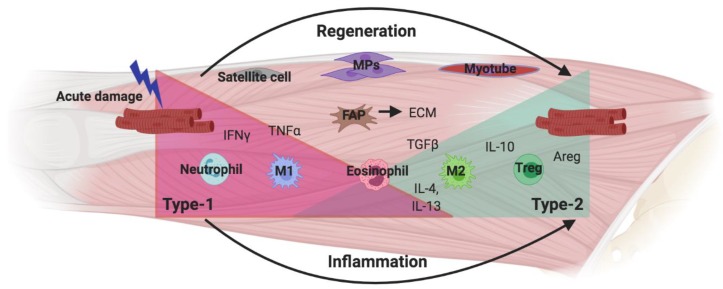
Skeletal muscle regeneration. Acute muscle fiber damage triggers pro-inflammatory immune cells to enter the muscle while satellite cells start to proliferate and generate a pool of muscle progenitors (MPs). Fibro/Adipo progenitors (FAPs) also proliferate and replace the broken extracellular matrix (ECM) during regeneration. A shift from a type-1 inflammatory environment to a type-2 anti-inflammatory environment allows for MP differentiation into muscle fibers.

**Table 1 ijms-21-01350-t001:** Innate lymphoid cell (ILC) subsets. The stimuli, cytokines and transcription factors (TFs) are summarized for each subset.

ILC Subsets		Stimuli	Cytokines	TFs
**ILC1**	ILC1	Intracellular infections, IL-12, IL-18	TNFα, IFNγ	Tbet
	NKs		TNFα, IFNγ Perforin, Granzymes	Tbet, EOMES
ILC2		Large parasites, tissue damage, IL-33, IL-25, TSLP	IL-4, IL-5, IL-9, IL-13, Areg	GATA3, RORα
ILC3	NCR+	Extracellular infections, TGFβ, IL-1β, IL-23	IL-22	RORγt
	NCR−		IL-22, IL-17	
	LTi			

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
