# Peer review of "Group 2 Innate Lymphoid Cells: Central Players in a Recurring Theme of Repair and Regeneration"

_ijms, 2020, doi:10.3390/ijms21041350_

Round 1

Reviewer 1 Report

In their review “Group 2 innate lymphoid cells: central players in a 2 recurring theme of repair and regeneration” authors overview the current knowledge about ILCs, particularly ILC2, including their phenotype and function, interaction with pro- and anti-inflammatory immune and stromal cell subsets, their role in the tissue homeostasis, inflammation, regeneration, fibrosis and neonatal immunity

The review is well written, has very clear illustrations and very effectively introduces and discusses complex biology of ILC2 for International Journal of Molecular Sciences readership.

Small point: it would be relevant to insert citations of studies that describe IL-17-producing ILCs in men (page 2 lines 60, 65 and Table 1).

Reviewer 2 Report

This review focuses on the interaction of ILC2s with stromal cells, eosinophils, macrophages, and T regulatory cells and how these interactions regulate inflammation, immunity, and fibrosis. They go on to propose how ILC2s might mediate skeletal muscle tissue regeneration.

This is a well-written, well-organized, review article. The figures are quite professional and enhance the manuscript.

I have several specific suggestions for improvement:

I think one cell interaction with ILC2s that is missing from this review is that of ILC2s and epithelial cells. In the lung ILC2s have been shown to improve epithelial restitution after viral injury (Monticelli, Nature Immunology 2011) and after intestinal epithelial injury (Monticelli, PNAS 2015) via Areg production. They also regulate epithelial secretory cell differentiation in the intestine (von Moltke, Nature 2015, Waddell, J Immunol 2019) via IL13 which is important for the weep and sweep response to parasites and potentially in response to inflammation. These mechanisms are equally important to others discussed when focusing on tissue repair and I recommend attention in the manuscript and figures. Page 1, line 33, would be important to add that a key distinction between ILCs and T cells is the lack of a rearranged T cell receptor and antigen specificity in ILCs Page 2, line 16, While GATA3 is a canonical Th2 cell transcription factor, to my knowledge RORalpha has not been established as such in Th2 cells. Section 2.4, T reg cells, it is in the figure, but probably worth discussing how IL-33 can directly activate T reg proliferation and suppressive function (Schiering, Nature, 2014) Page 8, line 291 – Others have proposed that IL-13 induces fibrosis through the IL13Ra2 and that this IS TGFbeta dependent (Fichtner-Feigl S, Gastroenterology 2008 and J Immunol 2007).
